

# A Study on the Key Factors Determining the Hygroscopic property of Black Carbon

Zhanyu Su[1,2], Lanxiadi Chen[2,3], Yuan Liu[1,2], Peng Zhang[1], Tianzeng Chen[1], Biwu Chu[1,2], Mingjin Tang[2,3], Qingxin Ma[1,2*], Hong He[1,2]

[1]State Key Joint Laboratory of Environment Simulation and Pollution Control, Research Center for Eco-Environmental Sciences, Chinese Academy of Sciences, Beijing 100085, China

[2]College of Resources and Environment, University of Chinese Academy of Sciences, Beijing 100049, China

[3]State Key Laboratory of Organic Geochemistry, Guangzhou Institute of Geochemistry, Chinese Academy of Sciences, Guangdong 510640, China

*Correspondence to*: Qingxin Ma (qxma@rcees.ac.cn)

**Abstract.** Black carbon (BC) is a crucial component of aerosols in the atmosphere. Understanding the hygroscopicity of BC particles is important for studying their role as cloud condensation nuclei (CCN) and ice nuclei (IN), as well as their chemical behavior and atmospheric lifetime. However, there is still a lack of comprehensive understanding regarding the factors that determine the hygroscopic properties of fresh BC. In this work, the hygroscopic behavior of BC particles generated from different types of fuel and aged with $SO_2$ for varying durations were measured by a vapor sorption analyzer while various characterizations of BC were conducted to understand the key factors that influence the hygroscopic properties of BC. It was found that the presence of water-soluble substances in BC facilitates the completion of monolayer water adsorption at low relative humidity, while also increasing the number of water adsorption layers at high relative humidity. On the other hand, BC prepared by burning organic fuels, which typically lacks water-soluble inorganic ions, primarily exhibits hygroscopicity characteristics influenced by organic carbon (OC) and microstructure. Furthermore, the hygroscopicity of BC can be enhanced by the formation of sulfate ions due to heterogeneous oxidation of $SO_2$. This finding sheds light on the critical factors that affect BC hygroscopicity during water adsorption and allows for estimating the interaction between water molecules and BC particles in a humid atmosphere.

**Keywords:** black carbon, hygroscopicity, multilayer adsorption, water-soluble ions, organic carbon, microstructure, heterogeneous reaction



**Introduction**

Black carbon (BC) particles are produced by incomplete combustion processes of carbon-containing

materials (Nie et al., 2020; Wei et al., 2020). The current global emission of BC has been estimated to

be 3-8 TgC per year (Forster et al., 2007). BC aerosol can influence climate by directly absorbing solar

radiation and affecting cloud formation and surface albedo through deposition on snow and ice (Liao et

al., 2015; Peng et al., 2016), which results in the contribution of BC to global warming second only to

that of $CO_2$ (Jacobson, 2001). In addition, BC particles can significantly enhance the atmospheric

oxidation capacity (He et al., 2022) and contribute to the formation of secondary aerosols by providing

active surface for the heterogeneous reactions of gaseous pollutants like $NO_2$, $SO_2$, and volatile organic

compounds (VOCs) (Tritscher et al., 2011; Han et al., 2017; Zhang et al., 2022b; Liu et al., 2023).

Moreover, BC particles also pose a health risk by causing and enhancing respiratory, cardiovascular,

and allergic diseases (Janssen et al., 2011; Lin et al., 2011). Due to its significant effect on global

climate change, regional air quality and human health, the physicochemical properties of BC have

attracted much attention in recent decades.

Hygroscopicity is one of the most important physicochemical properties of BC, which largely

determines the cloud condensation nuclei (CCN) and ice nucleation (IN) activity as well as the

consequent radiation forcing (Semeniuk et al., 2007; Ramanathan and Carmichael, 2008; Friedman et

al., 2011). On the other hand, the hygroscopicity of atmospheric particles is important for their

chemical behavior because water molecules were found to significantly affect the heterogeneous

transformation of gaseous pollutants on BC surfaces (Zhao et al., 2017; He and He, 2020; Zhang et al.,

2022b).

The hygroscopic behavior of BC has been widely studied. It was found that fresh BC or commercial

BC appears to be hydrophobic as there is no noticeable uptake of water at unsaturated humidity. For

instance, the commercial BC and spark discharge BC particles shrunk with increasing RH during the

growth factor measurements by hygroscopicity tandem differential mobility analysers (H-TDMA)

(Weingartner et al., 1997; Henning et al., 2010). This was explained with a restructuring of the

agglomerated particles. Due to the inverse Kelvin effect, water condenses in small angle cavities of BC

particles, which leads to capillary forces on the branches of the aggregates and cause them to collapse.

Different from the commercial BC and spark discharge BC, diesel BC, aircraft BC and biomass smoke



particle showed obvious particle size growth with increasing RH (Popovicheva et al., 2008; Carrico et al., 2010). This indicates that the chemical composition of BC is an important factor affecting its

hygroscopicity. Our previous study suggested that combustion conditions could affect morphology and microstructure of BC, which has significant effect on the hygroscopicity (Han et al., 2012).

BC aerosols experience internal mixing with non-BC compounds (inorganic, organic, or inorganic/organic mixtures) as aging after their emission (Shiraiwa et al., 2007; Matsui et al., 2013). Field observations have demonstrated that the presence of BC-coating materials greatly influences both

the hygroscopic properties and the CCN properties (or the wet removal) (Ohata et al., 2016; Li et al., 2018; Hu et al., 2021). Several laboratory studies have also simulated the hygroscopic changes of BC particles during atmospheric transport and aging. BC particles generated from incomplete combustion of propane were exposed to the oxidation products of the OH-toluene reaction, resulting in an organic coating that increased the hygroscopicity of the particles (Qiu et al., 2012). Moreover, the aging

process of propane flame BC through $NO_2$ oxidation of $SO_2$ was found to produce inorganic hydrophilic coating materials and significantly enhance the CCN activity of BC particles (Zhang et al., 2022a).

The hygroscopicity of BC can vary significantly depending on its source and aging processes, which has implications for regional air quality and climate. However, previous studies have often focused on

specific factors influencing the hygroscopicity of a particular type of BC, lacking a comprehensive understanding of the key factors determining the hygroscopic properties of fresh BC. In this study, we conducted measurements to determine the hygroscopicity of BC produced from different fuels and aged with $SO_2$ for different time. In addition, the chemical composition and microstructure of BC were characterized for each BC sample. The main objectives of this study were to compare the

hygroscopicity of BC from different sources and analyze the effect of OC, water-soluble ions and microstructure on the multilayer adsorption of BC surface water. Moreover, the impact of heterogeneous aging reactions on the multilayer adsorption of water on the surface of BC particles was also explored. This study contributes to a deeper understanding of the hygroscopicity and atmospheric impacts of BC particles in the atmosphere.



## 2.Experimental section

### 2.1 Black carbon Production.

Prepared BC particles were obtained by burning n-hexane, decane and toluene (AR, Sinopharm Chemical Reagent Lo., Ltd) in a co-flow system as described in our previous studies (Han et al., 2012; Zhao et al., 2017). Diesel black carbon (DBC) was collected from the diesel particle filter (DPF) of a China VI heavy-duty diesel engine (ISUZU from China). Printex U black carbon (UBC) from Degussa (CAS No.: 1333-86-4) was used as a model BC. These types of BC are usually used in laboratory simulation as representative of BC in the atmosphere (Liu et al., 2010; Han et al., 2012; Zhang et al., 2022b).

The aging experiments were performed in a quartz flow tube reactor. UBC was placed into a quartz flow tube reactor. The experiments were maintained at 298 K. Zero air was used as the carrier gas, and the total flow tube rate introduced in the flow tube reactor was 700 ml min$^{-1}$. The SO$_2$ concentration was 5 ppm. The relative humidity (RH) was adjusted by varying the ratio of dry zero air to wet zero air at 50 % and measured by a RH sensor (Vaisala HMP110). To simulate solar irradiation, a high uniformity integrated xenon lamp (PLS-FX300HU, Beijing Perfectlight Technology Co., Ltd.) of 270 mW cm$^{-2}$ was used as the light source. Its visible spectrum ranges from 330 to 850 nm.

### 2.2 Characterization of black carbon

Raman spectra of BC were obtained with a Renishaw inVia Raman microscope system using a 532 nm excitation wavelength. The exposure time for each scan was 60s. Data were acquired and analyzed using Renishaw WiRE 5.4 software.

The content of OC was measured using a thermal-optical transmittance OC/EC analyzer (Sunset laboratory Inc., Forest Grove, OR) with modified NIOSH 5040 protocol and produced four OC fractions (OC1, OC2, OC3, and OC4 at 150, 250, 450, and 550 °C respectively), OP (pyrolysed organic) fraction (a pyrolyzed carbonaceous component determined when transmitted laser returned to its original intensity after the sample was exposed to oxygen), and three EC fractions (EC1, EC2, and EC3 at 550, 700, and 800 °C, respectively). OC is defined as OC1+OC2+OC3+OC4+OP and EC is defined as EC1+EC2+EC3-OP (Chow et al., 1993; Li et al., 2016).

The species of OC in BC were analyzed and identified via gas chromatography coupled with mass spectrometry (GC–MS, Agilent 6890–5973). 5 mg BC was first ultrasonically extracted for 10 min



using 10 ml of dichloromethane ($CH_2Cl_2$), which was filtered through a quartz sand filter. The obtained

supernatant liquid was subsequently concentrated using the $N_2$ blowing method for final analysis. The

gas chromatograph was equipped with a DB-5MS 30 m × 0.25 mm × 0.25 mm capillary column and

the mass spectrometer employed a quadrupole mass filter with a 70eV electron impact ionizer. The

temperature of the programmed temperature vaporizer was held at 270 °C. The initial oven temperature

was set at 40 °C for 2 min, then increased step-by-step to 150 °C (by 5 °C min$^{-1}$) for 5 min, 280 °C (by

10 °C min$^{-1}$) for 10 min, and 320 °C (by 10 °C min$^{-1}$) for 5 min.

For ion chromatography (IC) measurement, about 5 mg of BC particles were extracted by

ultrasonication with 10 mL ultrapure water (specific resistance ≥ 18.2 MΩ cm) for 10 min. Then, the

extract was filtered through a 0.22 mm PTFE membrane filter. The obtained solution was analyzed

using a Wayee IC-6200 ion chromatography system equipped with a SI-524E anionic analytical

column. An eluent of 10 mM KOH was used at a flow rate of 1.0 mL min$^{-1}$.

**2.3 Hygroscopic properties of black carbon**

The hygroscopic properties of BC were investigated using a vapor sorption analyzer (VSA, Q5000 SA,

TA Instruments), which has been applied to study hygroscopicity of atmospherically relevant particles

in previous work (Chen et al., 2019; Gu et al., 2017). VSA utilizes a highly sensitive balance to

measure the mass change of a sample as a function of RH at a given temperature. The instrument has a

measurement range of 0−100 mg with a sensitivity of 0.01 μg, allowing for precise analysis. The

temperature could be controlled in the range of 5−85 °C with an accuracy of 0.1 °C, and RH could be

regulated in the range of 0−98 % with an absolute accuracy of 1 %. To ensure the accuracy of RH

measurements, we routinely measured deliquescence relative humidities (DRHs) of NaCl, $(NH_4)_2SO_4$,

and KCl, and the difference between measured and theoretical DRHs did not exceed 1 %, confirming

the reliability and accuracy of the instrument.

Hygroscopicity of BC was investigated at 298K. Figure 1 displays the change of RH and normalized

sample mass with experimental time in a typical experiment. UBC was dried at <1 % RH and the

sample mass under dry conditions was typically 1−5 mg. After that, RH was increased to 90 % start at

10 % step by step, and at each step, RH was increased by 20 %; at each RH, the sample was considered

to reach an equilibrium when its mass change was <0.05 % within 60 min, and then RH was changed to

the next value.



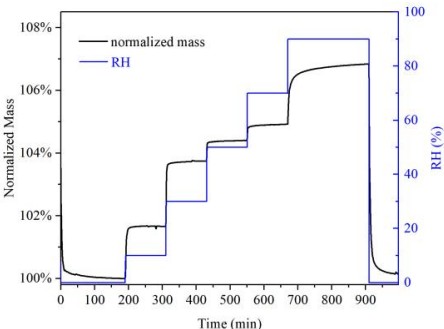

**Figure 1 RH (blue curve, right y axis) and normalized sample mass (black curve, left y axis) as a function of**
**experimental time during one experiment in which hygroscopic properties of UBC were examined at 298K.**

### 3.Result and discussion

### 3.1The vapor adsorption isotherms of various black carbon

Figure 2 shows the normalized sample mass (normalized to that at <1 % RH, $m/m_0$) as a function of
RH for five kinds of BC. Three types of prepared BCs (n-hexane BC, decane BC and toluene BC)
exhibited lower water adsorption per unit mass sample under each RH condition compared to DBC and
UBC particles. Specifically, at 90 % RH, DBC showed the highest water adsorption of all BC types,
with a $m/m_0$ value of 1.138, followed by UBC with a value of 1.067. Moreover, among the three
prepared BCs, decane flame BC exhibited the highest hygroscopicity, as indicated by its normalized
sample mass of 1.054 at 90 % RH.



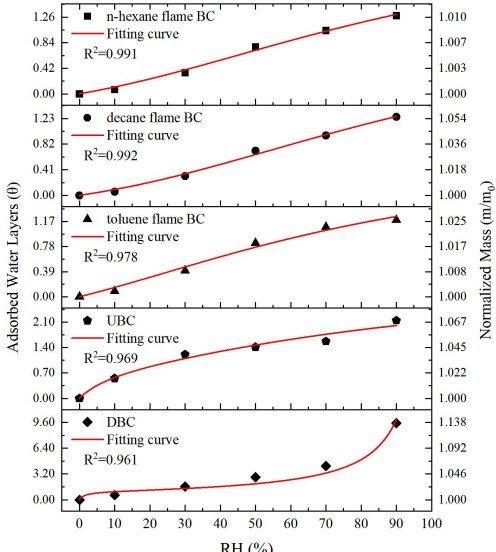


**Figure 2. Water adsorption isotherms of BCs, fitting curves (lines) with BET equation and the measured sample mass change (normalized to that at <1 % RH, i.e., m/m$_0$) of BCs as a function of RH (up to 90 % RH).**

In order to further analyze the adsorption characteristics of water on BC, the isotherms of BC were

fitted with the Brunauer-Emmett-Teller (BET) equation. As shown in Fig. 2, the isotherms of prepared

BC and UBC could be well fitted with three-parameters BET equation with the assumption of limited

adsorbed water layers as following Eq. (1)(Brunauer et al., 1938; Goodman et al., 2001; Ma et al., 2010;

Tang et al., 2016):

$$V = \frac{V_m c \frac{P}{P_0}}{1 - \frac{P}{P_0}} \times \frac{1-(n+1)\left(\frac{P}{P_0}\right)^n + n\left(\frac{P}{P_0}\right)^{n+1}}{1+(c-1)\frac{P}{P_0} - c\left(\frac{P}{P_0}\right)^{n+1}} \qquad (1)$$

where, $V$ is the volume of gas adsorbed at equilibrium pressure $P$, $V_m$ is the volume of gas necessary

to cover the surface of the adsorbent with a complete monolayer, $P$ is the equilibrium pressure of the

adsorbing gas, and $P_0$ is the saturation vapor pressure of the adsorbing gas at that temperature. $n$ is

an adjustable parameter given as the maximum number of layers of the adsorbing gas and is related to

the pore size and properties of adsorbent. As a result, multilayer formation of adsorbing gas is limited

to n layers at large values of $P/P_0$. The parameter $c$ is the temperature-dependent constant related to

the enthalpies of adsorption of the first and higher layers through Eq. (2) (Brunauer et al., 1938):



$$c = \exp\left(\frac{\Delta H_2^0 - \Delta H_1^0}{RT}\right) \tag{2}$$

where, $\Delta H_1^0$ is the standard enthalpy of adsorption of the first layer, and $\Delta H_2^0$ is the standard enthalpy of adsorption on subsequent layers and is taken as the standard enthalpy of condensation, $R$ is the gas

constant, and $T$ is the temperature in Kelvin.

For DBC, a notable increase in sample mass was observed between 70 % and 90 % RH. This can be attributed to a significant rise in the number of adsorbed water layers within this specific RH range. This hygroscopic characteristic of DBC particles leads to an adsorption isotherm that cannot be adequately described by the three-parameter BET equation, which assumes a limited number of

adsorbed water layers. However, the two-parameter BET equation (Eq. (3)), assuming an unlimited number of adsorbed water layers, provides a better fit for the observed adsorption behavior of DBC particles (Brunauer et al., 1938):

$$v = \frac{v_m c p}{(p_0 - p)\{1 + (c-1)(p/p_0)\}} \tag{3}$$

The fitted parameters, as shown in Table 1, provide valuable insights into the water adsorption

behavior of different BC. The threshold relative humidity for one monolayer (MRH) for the fresh prepared BCs is approximately 70 %. However, both UBC and DBC exhibit significantly lower MRH values ($MRH_{DBC}$=15 %, $MRH_{UBC}$=25.5 %) compared to fresh prepared BC. This suggests that UBC and DBC have a higher affinity for water uptake at lower RH levels compared to fresh BC. At 90 % RH, prepared BC and UBC particles were found to have approximately 1.2 and 2.1 layers of surface

water adsorbed, respectively. Interestingly, DBC showed a significantly higher number of surface water layers with around 9.5 layers adsorbed at 90 % RH. This indicates that DBC has a strong propensity for water adsorption and can accommodate a larger amount of adsorbed water compared to the other BC types.

The water-soluble ions like $SO_4^{2-}$ and $NO_3^-$ in BC samples were analyzed by IC, and the corresponding

results are presented in Table 2. It was observed that the content of $NO_3^-$ in all BC samples, except for DBC, was approximately 0.2 μg mg$^{-1}$. However, DBC exhibited a higher $NO_3^-$ content of 1.44 μg mg$^{-1}$, which could be due to the aging of high concentration NOx coexisting in the exhaust pipe. Regarding $SO_4^{2-}$, the fresh prepared BCs did not show any detectable levels of $SO_4^{2-}$. In contrast, both UBC and DBC displayed notable amounts of $SO_4^{2-}$, with UBC having a content of 2.54 μg mg$^{-1}$ and DBC having



the highest content of 11.46 µg mg$^{-1}$. These results indicate that water-soluble inorganic ions (e.g.,

nitrates and sulfates) are dominant factor to enhance the hygroscopicity of BC, which is consistent with

previous studies (Carrico et al., 2010; Popovicheva et al., 2010), highlighting the dominance of

water-soluble inorganic ions in influencing the hygroscopic properties of BC.

**Table 1. Adsorption parameters for water uptake on BC.**

| Black carbon | BET area (m$^2$ g$^{-1}$) | MRH (%) | n | c | R$^2$ |
|---|---|---|---|---|---|
| n-hexane flame BC | 26.27 | 68 | 2.84 | 1.01 | 0.991 |
| decane flame BC | 147.36 | 72 | 2.92 | 0.83 | 0.992 |
| toluene flame BC | 70.97 | 67.2 | 2.47 | 1.37 | 0.978 |
| DBC | 47.93 | 15 | --- | 66.95 | 0.961 |
| UBC | 97.24 | 25.5 | 3.34 | 9.57 | 0.969 |
| UBC aged 2h | 10.67 | 24.4 | 3.42 | 10.67 | 0.973 |
| UBC aged 6h | 9.82 | 25 | 3.59 | 9.82 | 0.956 |
| UBC aged 10h | 8.52 | 26.2 | 3.82 | 8.58 | 0.944 |


**Table 2. Mass of SO$_4^{2-}$ and NO$_3^-$ on BCs measured by IC.**

| Black carbon | Mass of SO$_4^{2-}$ (µg mg$^{-1}$) | Mass of NO$_3^-$ (µg mg$^{-1}$) |
|---|---|---|
| n-hexane flame BC | 0 | 0.19 |
| decane flame BC | 0 | 0.22 |
| toluene flame BC | 0 | 0.18 |
| DBC | 11.46 | 1.44 |
| UBC | 2.55 | 0.24 |
| UBC aged 2h | 4.83 | 0.20 |
| UBC aged 6h | 7.14 | 0.19 |
| UBC aged 10h | 9.61 | 0.20 |





**3.2 The factors controlling the hygroscopic properties of prepared black carbon**

Compared with DBC and UBC, prepared BCs are more hydrophobic (Fig. 2) due to less water-soluble

inorganic ion contained (Table 2). However, there are still significant differences in the hygroscopic

behavior of BCs prepared from different fuels. In order to analyze the differences in the hygroscopicity

of different prepared BC, the relative content and species of OC and microstructure of BC were

characterized. Fig. 3 shows the ratio of OC/EC of BC samples. It was found that the n-hexane flame

BC has the highest OC/EC ratio, followed by toluene flame BC, and decane flame BC has the lowest

OC/EC ratio. It should be noted that the ratio of OC/EC is negatively correlated with their

hygroscopicity, indicating that organic carbon is not conducive to the adsorption of water on the

surface of BC. The impact of OC on the hygroscopicity of BC is still a subject of debate. Some field

observations results have indicated that particles with high OC/EC ratio were preferentially removed by

precipitation and the condensation of photochemically generated secondary organic carbon on BC

particles could cause enhancement of hygroscopicity (Dasch and Cadle, 1989; Li et al., 2018).

However, HTDMA measurements have shown that neither hygroscopicity nor droplet activation of the

fresh propane BC particles depend on the OC content (Henning et al., 2012). Therefore, it is necessary

to analyze the specific OC species present in prepared BC particles to gain a better understanding of

their role in hygroscopicity.

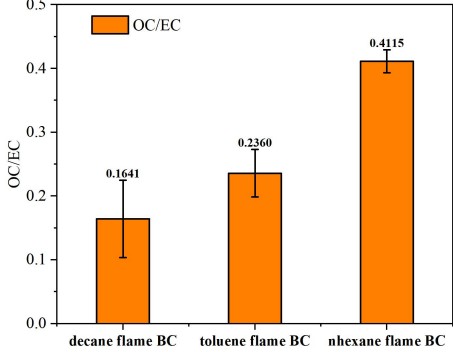


**Figure 3. the ratio of OC/EC of prepared BC measured by a thermal-optical transmittance**



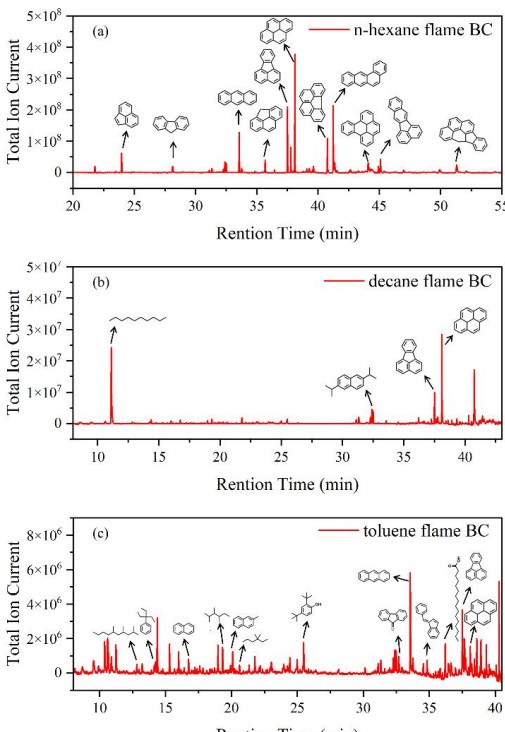

**Figure 4. Total ion chromatogram extracts of prepared BCs. (a) n-hexane flame BC. (b) decane flame BC. (c) toluene flame BC.**

In order to obtain the composition of OC in different BC, the samples were extracted by $CH_2Cl_2$ and the extract was analyzed by GC-MS. Fig. 4 shows the GC-MS analysis of OC extracted from different prepared BC. The major components are polyaromatic hydrocarbons (PAHs) like Anthracene, Fluoranthene and Pyrene in all BC samples. It is well known that PAHs are usually formed simultaneously with BC during prepared. In n-hexane flame BC, PAHs are the main OC while other

components are scarce, which is consistent with the results of Han et al. (Han et al., 2012). For decane and toluene flame BC, long-chain alkanes such as decane or 2,3,4-trimethyl-hexane are present in the OC fraction. The characteristic features and peculiarities of the adsorption of water vapor on BC are primarily caused by the tendency of polar water molecules to form hydrogen bonds (Vartapetyan and Voloshchuk, 1995). However, PAHs are weakly polar organic compounds. Thus, the ability of

$\pi$-electrons in the PAH aromatic ring to form weak hydrogen bonds with water molecules or the interactions with water molecules are either almost absent or are negligible (Lobunez, 1960). The




oxygen-containing functional groups of organic compounds are substantial centers for the formation of hydrogen bonds with water molecules. In toluene flame BC, certain OC compounds (2,4-Di-tert-butylphenol, palmitic acid and 9-fluorenone) possess oxygen-containing functional groups

like hydroxyl groups, carboxyl groups and quinone. However, these OC compounds also contain substantial hydrophobic parts (aromatic ring and hydrocarbon part). The contribution of these hydrophobic part of a molecule, which is quite small, to the hygroscopicity is dominant (Kireeva et al., 2010). For long-chain alkanes found in decane and toluene flame BC, composed solely of carbon and hydrogen atoms, they are typically considered hydrophobic. In general, OC constituents detected in

these prepared BC samples could impede water adsorption on BC surfaces. Hence, the presence of these OC compounds leads to hydrophobic characteristics and diminishes the water adsorption capacity of prepared BC.

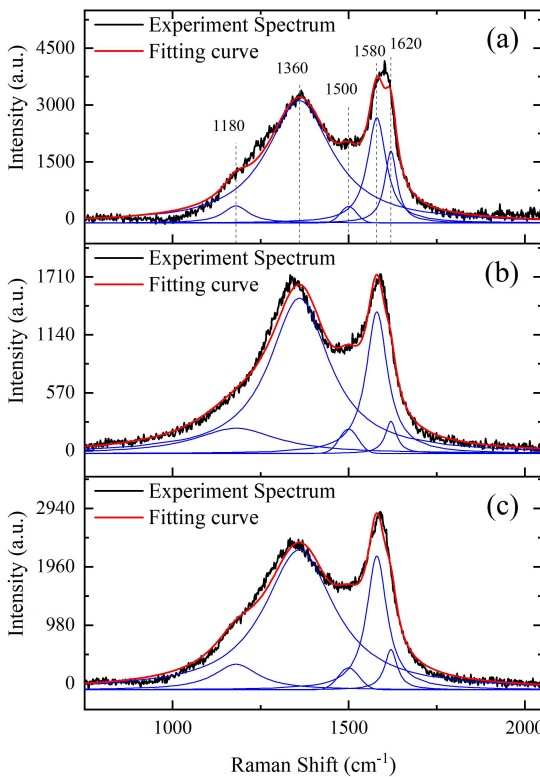

**Figure 5. Raman spectra of (a) n-hexane, (b) decane and (c) toluene flame BC.**



In order to study the relationship between the microstructure and vapor adsorption capacity of BC, Raman analysis on BC samples were conducted. Figure 5 shows the first-order Raman spectra of three prepared BCs with good curve-fitting results ($R^2 > 0.982$), which display well known bands of BC near 1580 (G band) and 1360 cm$^{-1}$ (D band). The G band is a typical characteristic of crystalline graphite, while the D band is only observed for disordered graphite. A detailed analysis of the first-order Raman

spectra was performed using the five-band fitting procedure proposed by Sadezky (Sadezky et al., 2005). Four Lorentzian-shaped bands (D1, D2, D4, and G, centered at about 1360, 1620, 1180, and 1580 cm$^{-1}$, respectively) and one Gaussian-shaped band (D3, centered at around 1500 cm$^{-1}$) were used in the curve-fitting process (Sadezky et al., 2005; Ivleva et al., 2007; Liu et al., 2010). The D1 band arises from the $A_{1g}$ symmetry mode of the disordered graphitic lattice located at the graphene layer

edges. The D2 band is attributed to the $E_{2g}$ symmetry stretching mode of the disordered graphitic lattice located at surface graphene layers. The D3 band originates from the amorphous carbon fraction of BC. The D4 band is related to the $A_{1g}$ symmetry mode of the disordered graphitic lattice or C−C and C=C stretching vibrations of polyene-like structures, polyenes and ionic impurities also contribute to the D4 band (Sze et al., 2001; Sadezky et al., 2005). The G band is assigned to the ideal graphitic lattice with

$E_{2g}$ symmetry vibration mode. The integral intensity ratio ($I_D/I_G$) of D and G bands could reflect the comparative content of disordered carbon at graphene layer edges and surface graphene layers, and the intensities of D and G bands have been widely determined using the sum of D1 and D4 bands and the sum of D2 and G bands. Table 3 shows that the $I_D/I_G$ of three prepared BCs have positive correlation with their hygroscopicity. These results imply that disordered graphitic lattice (D1), graphitic lattice,

polyenes, or ionic impurities (D4) could be favorable for the adsorption of water on BC.

**Table 3. Parameters $I_{D1}/I_G$, $I_{D2}/I_G$, $I_{D4}/I_G$ and $I_D/I_G$ of n-hexane, decane and toluene flame BC**

| Fuels | $I_{D1}/I_G$ | $I_{D4}/I_G$ | $I_{D2}/I_G$ | $I_D/I_G$ |
|---|---|---|---|---|
| n-hexane | 3.82±0.13 | 0.49±0.02 | 0.48±0.06 | 2.87±0.13 |
| toluene | 3.38±0.05 | 0.50±0.11 | 0.17±0.08 | 3.23±0.09 |
| decane | 3.17±0.30 | 0.85±0.01 | 0.17±0.08 | 3.49±0.15 |



### 3.3 The effect of aging process on the hygroscopicity of black carbon.

Based on the hygroscopicity measurements of UBC and DBC (Fig. 2 and 3, Table 2), it is evident that

the presence of water-soluble inorganic ions (e.g., sulfates and nitrates) can enhance the hygroscopicity

of BC particles. Field observations have shown that the mass fractions of ammonium, sulfate, and

nitrate increase with the aging of fresh biomass burning particles (Pratt et al., 2011). To investigate the

impact of sulfate formation during the aging process on BC hygroscopicity, we aged UBC with $SO_2$ for

different durations and measured their hygroscopic properties accordingly. The results revealed an

increase in sulfate ions on UBC with longer aging times (Table 2) while the MRH of UBC remains

relatively unchanged with $SO_2$ aging (Table 1). However, at 90 % RH, the adsorbed water layers on

UBC increases with increasing aging times (Fig. 6).

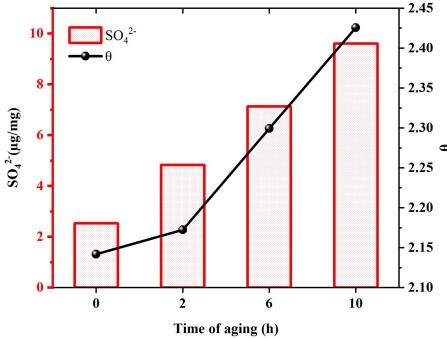

**Figure 6. Amounts of sulfates on UBC and the adsorbed water layers (θ) at 90 % RH of UBC as a function**
**of the time of aging.**

Our previous studies have also indicated that the heterogeneous reaction between $SO_2$ and BC leads to

the formation of sulfuric acid coating on the surface of BC (Zhang et al., 2022b). In this study, the

sulfate detected on UBC could also exist in the form of sulfuric acid. However, IC results demonstrated

that the amount of newly generated sulfate on UBC after 12 hours of aging was only 0.706 % of its

original mass (Table 2). This small amount is insufficient to cause a significant difference in mass

growth at low relative humidity which cause little change in MRH. However, Kireeva et al. showed

that the water adsorption isotherm of graphitized thermal soot coated with a small quantity of sulfuric

acid showed a significant increase in the mass growth factor slope of the coated soot at relative

humidity levels above 90 % (Kireeva et al., 2010). Zhang et al. found that coating with sulfuric acid

could increase the growth factor of BC to above 1.2 at 80 % RH relative to fresh particles (Zhang et al.,

2008). Our results also demonstrated that a noticeable augmentation in the amount of water adsorbed on SO₂ aged BC at high relative humidity levels. Based on these findings, it can be concluded that varying amounts of sulfuric acid produced through heterogeneous oxidation on the surface of BC lead to noticeable differences in the amount of adsorbed water at high relative humidity levels. These

findings are consistent with previous studies demonstrating that coating with sulfuric acid increases the hygroscopicity and ice nucleation activation of BC (Demott et al., 1999; Möhler et al., 2005; Wyslouzil et al., 1994).

### 4.Conclusion

In this study, we employed a vapor sorption analyzer to investigate the hygroscopicity of BC particles

from different sources and at different stages of aging with sulfur dioxide. Multiple characterizations of BC particles were also performed. DBC and UBC contained water-soluble ions, such as sulfates and nitrates, which enabled them to undergo monolayer adsorption at lower relative humidity and increase the number of water absorption layers at higher relative humidity. In contrast, fresh prepared BC particles, which have negligible amounts of water-soluble ions, were more hydrophobic. Their

hygroscopicity mainly depended on the organic carbon content and microstructure. A lower content of hydrophobic OC and a more disordered graphitic lattice, graphitic lattice, polyenes, or ionic impurities made prepared BC particles more prone to water adsorption. The aging of UBC particles with SO₂ resulted in the formation of water-soluble sulfate ions, which promotes an increase in the hygroscopicity of BC particles. This study analyzed the key factors determining the hygroscopic

property of BC, including water-soluble ions, organic carbon content, and microstructure. And provides a basis for improving our understanding of the hygroscopic behavior of sulfate-mixed BC in the atmosphere, which could help to evaluate changes in hygroscopicity during the heterogeneous reactions of BC particles with pollutant gases in future studies.

**Data availability**

The experimental data are available upon request to the first or corresponding authors

**Author Contributions**



QM contributed to the conception of the study, ZS and LC designed and conducted this experiment, YL
helped to prepare samples. QM, PZ, TC, BC, MT and HH helped perform the analysis with
constructive discussions. ZS and QM wrote the paper with input from all coauthors. All authors
contributed to the final paper.

**Competing interests**

The authors declare that they have no conflict of interest.

**Acknowledge**

This work was supported by the National Key R&D Program of China (2022YFC3701004), and the
National Natural Science Foundation of China (No. 22188102), The authors also appreciate the Youth
Innovation Promotion Association, CAS (Y2022021).

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
