# Peer review of "A study on the influence of inorganic ions, organic carbon, and microstructure on the hygroscopic property of soot"

_EGUsphere, 2023_

## Author Comment (AC1)

**Response to Reviewer 1:**

Review of "A Study on the Key Factors Determining the Hygroscopic property of Black Carbon" by Su et al.

In this study, the authors investigate the hygroscopicity of five types of black carbon (BC) samples in the laboratory by measuring the change in mass of each sample as a function of relative humidity. They also try to interpret the hygroscopicity data by analyzing BC samples with the OC/EC analyzer, GC-MS, IC, and Raman spectroscopy. Because the hygroscopicity of BC-containing particles is an important parameter affecting the atmospheric behavior and climatic impacts of BC, the data presented here is valuable. However, many clarifications are required so that the readers can understand their experimental procedures and the validity of their results and interpretation. The reviewer believes major revisions are necessary before its publication in ACP.

**Reply:** Thank you very much for the time and effort spent in reviewing the manuscript. Your guidance is greatly appreciated, and we believe it will significantly improve our manuscript. The following is a point-by-point response to all the comments, and the manuscript has been revised substantially according to your comments.

**Major comments**

**Q1.** Due to a lack of information and explanation, the reviewer could not understand the details of the experiments. For example, were the BC particles generated by the combustion of n-hexane, decane, and toluene collected on filters for further experiments? Is the U black carbon (UBC) sample originally suspension or a dry powder? No details of the aging experimental setup were also provided. Are there any previous studies using the same setup? What are the diesel engine operating conditions for collecting diesel BC (DBC) samples? It is difficult for the reader to reproduce their experiments with only the information provided in this manuscript.

**A1.** Thanks for your reminding. Here, we show the detail information about these experimental setups and added in the revised manuscript.

1. For prepared BC particles, they were obtained by burning n-hexane, decane and

toluene (AR, Sinopharm Chemical Reagent Lo., Ltd) in a co-flow system as described in our previous studies (Han et al., 2012; Zhao et al., 2017). Soot was collected on a quartz disc (7 cm in diameter) over a diffusion flame, eventually stored in a brown bottle (Agilent).

2. UBC (Printex U from Degussa, CAS No.: 1333-86-4) samples are dry powders and used as purchased. The aging experiments were performed in a quartz flow tube reactor (12 cm in length, 10 mm in diameter), as seen below. UBC was placed into a quartz flow tube reactor. $SO_2$ with certain concentration was introduced with carrier air. The RH in the flow could be established by adjusting the ratios of dry and humid air. A Xenon was used as light source of photo oxidation. This method has been used in our previous study (Zhang et al., 2022).

[Figure]

3. Diesel black carbon (DBC) was collected from the diesel particle filter (DPF) of a China VI heavy-duty diesel engine (ISUZU from China). A diesel engine bench test was run under the conditions of World Harmonized Transient Cycle (WHTC). China VI fuels were used in the study, meeting the GB T32859-2016 standard. The mechanism of $SO_2$ photooxidation on these DBC was studied in our previous study (Zhang et al., 2022).

This detailed information has been added in the revised manuscript (Line 88-102)

**Q2.** The definition of the term "BC" should be clearly stated. In this study, the authors seem to refer to a BC particle as an entire soot particle that can contain other organic or inorganic compounds. However, this treatment can make various descriptions unclear. For example, an expression like "OC in BC" (Line 112) would be confusing for many

researchers in the community. As discussed in previous studies (e.g., Petzold et al., 2013), it may be clear if mixed particles containing a BC fraction are termed "BC-containing particles" instead of "BC particles." Please consider the terminology used in this study and state it clearly.

**A2:** Thanks for your suggestion. Previous study (Petzold et al., 2013) have mentioned that soot denotes the ensemble of the particles emitted during incomplete combustion, i.e., the sum of black carbon and organic carbon. The term soot generally refers to the source mechanism of incomplete combustion of hydrocarbon fuels rather than to a material property. In this study, five types of BC particles were analyzed. Three of the black carbon were produced by the co-flow burner and fueled with n-hexane, decane, and toluene. Diesel black carbon was produced in a diesel engine with China VI fuels. All of them are products of incomplete combustion of hydrocarbon fuels, and Printex U powder was used as a model soot. Therefore, we think that "soot" is more appropriate than black carbon for samples used in this study.

Names of samples have been changed in the revised manuscript:

n-hexane / decane / toluene flame BC → n-hexane / decane / toluene flame soot

Diesel BC → diesel soot (DS)

Printex U BC → Printex U powder (U-soot).

All "black carbon" and "BC" was changed to "soot" in the revised manuscript.

**Q3.** In the reviewer's understanding, this study analyses bulk samples, which means that the information on the mixing states of the particles collected on filters (e.g., mixing states of the particles collected from diesel exhaust) is not obtained. Because this study discusses the hygroscopicity based on the increase in the mass of each sample, does it implicitly assume that all non-BC components are internally mixed with BC? In other words, is the measured hygroscopicity considered an upper limit of the hygroscopicity of BC-containing particles? (If some non-BC components are externally mixed with BC, then the hygroscopic growth (mass increase) of the non-BC component should not contribute to the hygroscopic growth of BC-containing particles.) Explanations should

be given.

**A3.** According to the previous comment (**Q2**), we have replaced black carbon to soot to describe the particles which contains EC, OC and other components. It is known that soot can exist in external mixing state (distinct from other components) and internal mixing state (incorporated within other components) (Jacobson, 2001). In this study, all soot samples are products of incomplete combustion of hydrocarbon fuels. During combustion processes, organic carbon (OC) is cogenerated with elemental carbon (EC) and mixed with EC through adsorption (Ogren and Charlson, 1983). Thus, OC and EC are usually assumed in an internal mixing state in soot particles.

As for inorganic components in soot, previous study demonstrated that diesel soot produced from diesel fuel contains up to 3.5% water-soluble substances, with the major fraction being comprised of inorganic ions such as sulfate ($SO_4^{2-}$) and chloride ($Cl^-$) (Lammel and Novakov, 1995). In our study, the IC results revealed the significant presence of sulfate ($SO_4^{2-}$) and nitrate ($NO_3^-$) ions within Diesel Soot particles. In order to clarify the mixing state of these water-soluble inorganic salts with DS, we performed TEM characterization with different dispersible liquids (ethanol and ultrapure water).

[Figure]

**Figure s1.** TEM images of diesel soot, (a) sample were ultrasonically dispersed in ethanol, (b) sample were ultrasonically dispersed in ultrapure water.

As seen in Figure s1a, the images of diesel soot aggregates dispersed in ethanol show a typical chain-like agglomerated structure composed of small spherical primary particles. In contrast, the samples dispersed in ultrapure water show many regular rectangle images due to salt crystallization (Figure s1b). Thus, it can be inferred that diesel soot and inorganic salts exist in a state of internal mixing.

As for $SO_2$ aged samples, soot particles and sulfate species could also exist in a

state of internal mixing, since our previous study demonstrated that $SO_2$ is oxidized by photo induced OH radicals on soot surface to form sulfuric acid (Zhang et al., 2022). Therefore, in our study, the mixing states of all soot particles could be considered in internal mixing state.

**Q4.** A clearer description of this study's new findings should be provided. The authors point out "a lack of comprehensive understanding regarding the factors that determine the hygroscopic properties of fresh BC" in the abstract and introduction (Lines 15 and 75). From that perspective, however, aging experiments with $SO_2$ are not intended to gain insight into fresh BC and are, therefore, not directly relevant. Also, it is well known that coatings of inorganic substances increase the hygroscopicity of BC-containing particles.

**A4.** Thanks for your comments and suggestions. The new findings in this study include quantitative measurement of vapor sorption capacity of various soot particles, the relationship between the hygroscopicity and the microstructure of soot, the effect of typical mixing components on the hygroscopicity of soot. As for fresh soot, we originally meant soot particles produced by fuel combustion, which is always considered to be hydrophobic. Because the hygroscopicity of soot plays an important role in its aging process, exploring the critical factors affecting the hygroscopicity of fresh produced soot is of significance for understanding their aging process. We intended to reveal the influence of microphysics and the content and composition of OC on the hygroscopicity of fresh soot particles. Nevertheless, we think that the use of the term "fresh" in the abstract and introduction sections is not necessary and could lead to misunderstandings. Therefore, we have deleted the word "fresh" from these sections. (Line 15 and 75).

We agree that it is well known that coatings of inorganic substances increase the hygroscopicity of BC-containing particles. Field studies showed that coating nitrate and sulfate on BC particles can increase the growth factor of these BC particles up to ~1.4-1.6 (Li et al., 2018; Liu et al., 2013). In laboratory investigations exploring the impact of sulfate on the hygroscopicity of BC, condensation of gaseous sulfuric acid has been

commonly used to simulate the mixing process of sulfate and BC in the atmosphere. Previous study showed that the hygroscopic growth factor of BC particles exposed to sulfuric acid vapor can reach 1.52 at 90% RH which is lower than pure sulfuric acid particles (2.03) (Zhang et al., 2008). Khalizov et al. found similar results with the hygroscopic growth factor of sulfuric acid coated soot particles in the range from 1.32 to 1.49 at 90% RH (Khalizov et al., 2009). These results represent the case where soot particles are suspended in the ambient atmospheric with high $H_2SO_4$ level, which may not suitable for most real atmosphere. Our previous study found that the photo oxidation of $SO_2$ on soot surface could be potential formation pathway to form internal mixture of sulfuric acid and soot (Zhang et al., 2022). Thus, in this study, we prepared $SO_2$ aged soot particles by the direct heterogeneous photochemical reaction between $SO_2$ and soot, which will promote the understanding of the hygroscopicity of soot particles containing sulfuric acid in the atmosphere.

**Q5.** The title is too broad and should be more specific. English proofreading throughout the manuscript is also desired.

**A5.** The title was changed to "A study on the influence of inorganic ions, organic carbon, and microstructure on the hygroscopic property of soot". We have also had native English speaker polish our English writing.

**Specific comments:**

**Q1.** L34, There have been many studies on the radiative effects of BC after 2001, and their understanding has been much updated. Please consider adding or updating the references.

**A1.** We have updated the reference in the revised manuscript (Line 35):

Soot aerosol can influence climate by directly absorbing solar radiation and affecting cloud formation and surface albedo through deposition on snow and ice (Liao et al., 2015; Peng et al., 2016), which results in the contribution of soot to anthropogenic radiative forcing second only to that of $CO_2$ (Bond et al., 2013; Cappa et al., 2012; Liu et al., 2017)

Bond, T. C., Doherty, S. J., Fahey, D. W., Forster, P. M., Berntsen, T., DeAngelo, B. J., Flanner, M. G., Ghan, S., Kaercher, B., Koch, D., Kinne, S., Kondo, Y., Quinn, P. K., Sarofim, M. C., Schultz, M. G., Schulz, M., Venkataraman, C., Zhang, H., Zhang, S., Bellouin, N., Guttikunda, S. K., Hopke, P. K., Jacobson, M. Z., Kaiser, J. W., Klimont, Z., Lohmann, U., Schwarz, J. P., Shindell, D., Storelvmo, T., Warren, S. G., and Zender, C. S.: Bounding the role of black carbon in the climate system: A scientific assessment, Journal of Geophysical Research-Atmospheres, 118, 5380-5552, 10.1002/jgrd.50171, 2013.

Cappa, C. D., Onasch, T. B., Massoli, P., Worsnop, D. R., Bates, T. S., Cross, E. S., Davidovits, P., Hakala, J., Hayden, K. L., Jobson, B. T., Kolesar, K. R., Lack, D. A., Lerner, B. M., Li, S.-M., Mellon, D., Nuaaman, I., Olfert, J. S., Petaja, T., Quinn, P. K., Song, C., Subramanian, R., Williams, E. J., and Zaveri, R. A.: Radiative Absorption Enhancements Due to the Mixing State of Atmospheric Black Carbon, Science, 337, 1078-1081, 10.1126/science.1223447, 2012.

Liu, D., Whitehead, J., Alfarra, M. R., Reyes-Villegas, E., Spracklen, D. V., Reddington, C. L., Kong, S., Williams, P. I., Ting, Y.-C., Haslett, S., Taylor, J. W., Flynn, M. J., Morgan, W. T., McFiggans, G., Coe, H., and Allan, J. D.: Black-carbon absorption enhancement in the atmosphere determined by particle mixing state, Nature Geoscience, 10, 184-U132, 10.1038/ngeo2901, 2017.

**Q2.** L51, Is "commercial BC" not "fresh BC"? Is the UBC a commercial BC in this study?

**A2.** In this sentence, "fresh BC" means soot prepared by fuel combustion in the laboratory, which was used for comparison with aged samples. UBC is a kind of commercial BC (Printex U from Degussa, CAS No.: 1333-86-4). The ion chromatography analysis results indicate that UBC contains a small amount of sulfate, which may be due to aging during transportation and storage. Therefore, we think UBC could not be considered fresh BC. The corrections have been made in the manuscript (Line 49-51):

The hygroscopic behavior of soot has been widely studied. It was found that soot prepared in the laboratory or commercial soot appears to be hydrophobic as there is no noticeable uptake of water at unsaturated humidity.

**Q3.** L111, EC = EC1+EC2 + EC3?

**A3.** When the content of OC and EC was measured using a thermal-optical transmittance OC/EC analyzer, OC is usually defined as OC1+OC2+OC3+OC4+OP and EC is defined as EC1+EC2+EC3-OP (Chow et al., 1993; Li et al., 2016). The reason is that the EC fraction formed by OC conversion during pyrolysis is referred to as pyrolyzed carbon (OP) (Boparai et al, 2008). Thus, OP should be subtracted from the EC fraction and added to the OC fraction.

**Q4.** L166: P has already been defined in L165.

**A4.** The sentence "P is the equilibrium pressure of the adsorbing gas" has been deleted from the revised manuscript.

**Q5.** L183: v and v_m are lowercase letters in Eq. (3) but uppercase letters in Eq. (1).

**A5.** Thank you for your careful check. The corrections have been made in the manuscript (Line 201):

$$V = \frac{V_m c P}{(P_0 - P)\{1 + (c - 1)(P/P_0)\}}$$

**Q6.** L195: Table 2 lists the concentrations of $SO_4^{2-}$ and $NO_3^-$. Can the mass of EC and OC also be listed in the same Table?

**A6.** The ratio of OC/EC of soot samples has been added in the Table 2. (Line 221):

**Table 2. Mass concentration of $SO_4^{2-}$ and $NO_3^-$ and the ratio of OC/EC of soots.**

| Soot | Mass concentration of $SO_4^{2-}$ ($\mu g\ mg^{-1}$) | Mass concentration of $NO_3^-$ ($\mu g\ mg^{-1}$) | OC/EC |
|---|---|---|---|
| n-hexane flame soot | 0.00 | 0.19 | 0.41±0.02 |

| | | | |
|---|---|---|---|
| toluene flame soot | 0.00 | 0.18 | 0.24±0.04 |
| decane flame soot | 0.00 | 0.22 | 0.16±0.06 |
| DS | 11.46 | 1.44 | 0.14±0.02 |
| U-soot | 2.55 | 0.24 | 0.12±0.03 |
| U-soot aged 2h | 4.83 | 0.20 | --- |
| U-soot aged 6h | 7.14 | 0.19 | --- |
| U-soot aged 10h | 9.61 | 0.20 | --- |

**Q7.** L213: Five different samples were used in this study, but the results from the OC/EC, GC-MS, and Raman analyses are only shown for the three samples. Why is that?

**A7.** In section 3.1, the inorganic ions present in five different black carbons were analyzed using IC. The findings revealed that DBC and UBC contained significant amounts of $SO_4^{2-}$ and $NO_3^-$ ions. On the other hand, the three prepared black carbons exhibited minimal levels of inorganic ions. Thus, the three prepared black carbons were chosen to investigate the impact of factors other than inorganic ions on the hygroscopic properties of black carbon. Because inorganic components affect the hygroscopicity more significantly than OC and microstructure, the information of OC and microstructure of DBC and UBC was not shown.

**Q8.** L273: If I_D is the sum of I_D1 and I_D4, why are some values of I_D1/I_G larger than I_D/I_G in Table 3?

**A8.** Thank you for your careful check. We realized that there were some errors in the data handling process. As mentioned above, the first-order Raman spectra of three prepared soots has good curve-fitting results, four Lorentzian-shaped bands (D1, D2, D4, and G, centered at about 1360, 1620, 1180, and 1580 cm$^{-1}$, respectively) and one Gaussian-shaped band (D3, centered at around 1500 cm$^{-1}$) were used in the curve-fitting process. The intensities of G bands (IG) have been widely determined using the sum of D2 and G bands. The value of ID1/IG is the area of the D1 band divided by the sum of

the areas of the D2 and G bands. However, I incorrectly calculated the area of the D1 band divided by the area of the G band. The same error exists in the calculation of the value of ID4/IG and ID2/IG. Considering that the value of ID2/IG has no meaning in this paper, I have deleted it. I fixed the error and put the new result in the table 3 (Line 290):

Table 3. Parameters $I_{D1}/I_G$, $I_{D4}/I_G$, $I_D/I_G$ and $L_a$ of n-hexane, decane and toluene flame soot.

| Fuels | $I_{D1}/I_G$ | $I_{D4}/I_G$ | $I_D/I_G$ | $L_a$(Å) |
|---------|--------------|--------------|-----------|------------|
| n-hexane | 2.59±0.13 | 0.33±0.01 | 2.87±0.13 | 15.34±0.69 |
| toluene | 2.87±0.03 | 0.42±0.09 | 3.23±0.09 | 13.62±0.38 |
| decane | 2.69±0.08 | 0.75±0.01 | 3.49±0.15 | 12.63±0.53 |

**Q9.** L295: Under the experimental RH conditions lower than the deliquescence RH of ammonium sulfate, ammonium sulfate would not contribute to hygroscopic growth even if present in abundance?

**A9.** The IC experiment results of UBC show that it does not contain $NH_4^+$ cations. Our previous study showed that heterogeneous reaction between $SO_2$ and BC leads to the formation of sulfuric acid coating on the surface of BC (Zhang et al., 2022). The presence of sulfuric acid would not contribute to hygroscopic growth under low RH condition.

**Q10.** L306: Since this study examines not the ice nucleating property but the hygroscopicity of BC-containing particles, the comparison with previous studies focusing on the ice nucleating properties of BC may not be appropriate here. I suggest adding more explanation or eliminating the description of the ice nucleation. In addition, since it is now common to refer to ice nucleating particles (INPs) rather than IN, I suggest modifying or eliminating the use of the term IN in the abstract.

**A10.** Thanks for your suggestion. Due to its lack of relevance in the article, the mention of black carbon's ice nucleation property was deemed inappropriate and subsequently removed.

**REFERENCE**

Boparai, P., Lee, J., and Bond, T. C.: Revisiting Thermal-Optical Analyses of Carbonaceous Aerosol Using a Physical Model, Aerosol Science and Technology, 42, 930-948, 10.1080/02786820802360690, 2008.

Chow, J. C., Watson, J. G., Pritchett, L. C., Pierson, W. R., Frazier, C. A., and Purcell, R. G.: THE DRI THERMAL OPTICAL REFLECTANCE CARBON ANALYSIS SYSTEM - DESCRIPTION, EVALUATION AND APPLICATIONS IN UNITED-STATES AIR-QUALITY STUDIES, Atmospheric Environment Part a-General Topics, 27, 1185-1201, 10.1016/0960-1686(93)90245-t, 1993.

Han, C., Liu, Y., Liu, C., Ma, J., and He, H.: Influence of combustion conditions on hydrophilic properties and microstructure of flame soot, J Phys Chem A, 116, 4129-4136, 10.1021/jp301041w, 2012.

Jacobson, M. Z.: Strong radiative heating due to the mixing state of black carbon in atmospheric aerosols, Nature, 409, 695-697, 10.1038/35055518, 2001.

Khalizov, A. F., Zhang, R., Zhang, D., Xue, H., Pagels, J., and McMurry, P. H.: Formation of highly hygroscopic soot aerosols upon internal mixing with sulfuric acid vapor, Journal of Geophysical Research-Atmospheres, 114, 10.1029/2008jd010595, 2009.

Lammel, G. and Novakov, T.: WATER NUCLEATION PROPERTIES OF CARBON-BLACK AND DIESEL SOOT PARTICLES, Atmospheric Environment, 29, 813-823, 10.1016/1352-2310(94)00308-8, 1995.

Li, C., Hu, Y., Chen, J., Ma, Z., Ye, X., Yang, X., Wang, L., Wang, X., and Mellouki, A.: Physiochemical properties of carbonaceous aerosol from agricultural residue burning: Density, volatility, and hygroscopicity, Atmospheric Environment, 140, 94-105, 10.1016/j.atmosenv.2016.05.052, 2016.

Li, K., Ye, X., Pang, H., Lu, X., Chen, H., Wang, X., Yang, X., Chen, J., and Chen, Y.: Temporal variations in the hygroscopicity and mixing state of black carbon aerosols in a polluted megacity area, Atmospheric Chemistry and Physics, 18, 15201-15218, 10.5194/acp-18-15201-2018, 2018.

Liu, D., Allan, J., Whitehead, J., Young, D., Flynn, M., Coe, H., McFiggans, G., Fleming, Z. L., and Bandy, B.: Ambient black carbon particle hygroscopic properties controlled by mixing state and composition, Atmospheric Chemistry and Physics, 13, 2015-2029, 10.5194/acp-13-2015-2013, 2013.

Ogren, J. A. and Charlson, R. J.: ELEMENTAL CARBON IN THE ATMOSPHERE - CYCLE AND LIFETIME, Tellus Series B-Chemical and Physical Meteorology, 35, 241-254, 1983.

Petzold, A., Ogren, J. A., Fiebig, M., Laj, P., Li, S. M., Baltensperger, U., Holzer-Popp, T., Kinne, S., Pappalardo, G., Sugimoto, N., Wehrli, C., Wiedensohler, A., and Zhang, X. Y.: Recommendations for reporting "black carbon" measurements, Atmospheric Chemistry and Physics, 13, 8365-8379, 10.5194/acp-13-8365-2013, 2013.

Zhang, P., Chen, T., Ma, Q., Chu, B., Wang, Y., Mu, Y., Yu, Y., and He, H.: Diesel soot photooxidation enhances the heterogeneous formation of $H_2SO_4$, Nat Commun, 13, 5364, 10.1038/s41467-022-33120-3, 2022.

Zhang, R., Khalizov, A. F., Pagels, J., Zhang, D., Xue, H., and McMurry, P. H.: Variability in morphology, hygroscopicity, and optical properties of soot aerosols during atmospheric processing, Proceedings of the National Academy of Sciences of the United States of America, 105, 10291-10296, 10.1073/pnas.0804860105, 2008.

Zhao, Y., Liu, Y., Ma, J., Ma, Q., and He, H.: Heterogeneous reaction of $SO_2$ with soot: The roles of relative humidity and surface composition of soot in surface sulfate formation, Atmospheric Environment, 152, 465-476, 10.1016/j.atmosenv.2017.01.005, 2017.

---

## Author Comment (AC2)

**Response to Reviewer 2:**

**Comments:**

Black carbon is a crucial component of aerosols in the atmosphere and the corresponding hygroscopicity is important for studying their CCN, IN and lifetime properties. In this study, the hygroscopic properties of BC parties from different fuel and aging process were measured. The results is interesting and convincing. I recommend this manuscript to be published after some major revisions.

**Reply:** We would like to thank you for the time and effort spent in reviewing the manuscript. Your valuable comments are helpful for improve our manuscript. The following is a point-by-point response to all the comments, and the manuscript has been revised substantially according to your comments.

**Major Comments**

**Q1.** One of the major concerns is the basic microphysical properties of the generated BC particles from different types. Are there any size distribution and morphology information about the DBC and UBC? These properties are important for understanding the hygroscopicity of the BC.

**A1.** We added the results of TEM experiments on various soot samples and the TEM images of soot samples are shown below. All soot samples consisted of typical spherical particles, which formed long chainlike agglomerates as reported in other studies (Han et al., 2012; Liu et al., 2010). The diameter distribution of soot particles based on TEM analysis are also shown below. Particles exhibit a relatively uniform particle size distribution.

[Figure]

**Figure 2.** TEM images of n-hexane flame soot (A), decane flame soot (B), toluene flame soot (C), diesel soot (D), U-soot aggregates (E) before and (F) after aged with 5 ppm of SO$_2$ for 10 h.

[Figure]

**Figure 3.** Diameter distribution of n-hexane flame soot, decane flame soot, toluene flame soot and diesel soot and U-soot particles before and after aged with 5 ppm of SO$_2$ for 10 h.

These discussions have been added in the revised manuscript (Line 154-167).

**Q2.** The VSA measure the hygroscopic properties of the BC. I didn't get the information that the measured results represent the BC particles of bulk information or single

particle? The author mentioned that the mass of BC under dry conditions was typically 1-5 mg. Please give us the size information of BC particles as the size is a very important parameter that relate the hygroscopicity with the mass increment.

**A2.** The vapor sorption analyzer (VSA) has been introduced and applied to study hygroscopicity of atmospherically relevant particles in previous work (Gu et al., 2017). This instrument consists of two main parts: (1) a high-precision balance used to measure the mass of samples and (2) a humidity chamber in which temperature and RH can be precisely regulated and also monitored online. The VSA measurement results in a mass change during water absorption of the bulk black carbon powder. Size distribution and morphology information about the soot samples are showed in Figure 2 and 3. These discussions have been added in the revised manuscript (Line 154-167).

**Minor Comments:**

**Q1.** Maybe a table is enough for figure 3.

**A1.** Thanks for your suggestion. The ratio of OC/EC of soot samples has been listed in the Table 2 and the OC/EC figure is deleted. (Line 221):

Table 2. Mass concentration of $SO_4^{2-}$ and $NO_3^-$ and the ratio of OC/EC of soots.

| Soot | Mass concentration of $SO_4^{2-}$ ($\mu g\ mg^{-1}$) | Mass concentration of $NO_3^-$ ($\mu g\ mg^{-1}$) | OC/EC |
|---|---|---|---|
| n-hexane flame soot | 0.00 | 0.19 | 0.41±0.02 |
| toluene flame soot | 0.00 | 0.18 | 0.24±0.04 |
| decane flame soot | 0.00 | 0.22 | 0.16±0.06 |
| DS | 11.46 | 1.44 | 0.14±0.02 |
| U-soot | 2.55 | 0.24 | 0.12±0.03 |
| U-soot aged 2h | 4.83 | 0.20 | --- |
| U-soot aged 6h | 7.14 | 0.19 | --- |
| U-soot aged 10h | 9.61 | 0.20 | --- |

**Q2.** In section 3.1 there are five types of BC particles. Why are there only three type of

BC presented in section 3.2?

**A2.** In section 3.1, the inorganic ions present in five different black carbons were analyzed using IC. The findings revealed that DBC and UBC contained significant amounts of $SO_4^{2-}$ and $NO_3^-$ ions. On the other hand, the three prepared black carbons exhibited minimal levels of inorganic ions. Thus, the three prepared black carbons were chosen to investigate the impact of factors other than inorganic ions on the hygroscopic properties of black carbon. Because inorganic components affect the hygroscopicity more significantly than OC and microstructure, the information of OC and microstructure of DBC and UBC was not shown.

**Q3.** As best as I know, the hygroscopic properties of BC particles were not directly related with the ice nucleation activation of BC in this study. The author mentioned the ice nucleation activation for many times but I don't think they necessary.

**A3.** Thanks for your suggestion. Due to its lack of relevance in the article, the mention of black carbon's ice nucleation property was deemed inappropriate and subsequently removed.

**REFERENCE**

Gu, W., Li, Y., Zhu, J., Jia, X., Lin, Q., Zhang, G., Ding, X., Song, W., Bi, X., Wang, X., and Tang, M.: Investigation of water adsorption and hygroscopicity of atmospherically relevant particles using a commercial vapor sorption analyzer, Atmospheric Measurement Techniques, 10, 3821-3832, 10.5194/amt-10-3821-2017, 2017.

Han, C., Liu, Y., Liu, C., Ma, J., and He, H.: Influence of combustion conditions on hydrophilic properties and microstructure of flame soot, J Phys Chem A, 116, 4129-4136, 10.1021/jp301041w, 2012.

Liu, Y., Liu, C., Ma, J., Ma, Q., and He, H.: Structural and hygroscopic changes of soot during heterogeneous reaction with O-3, Physical Chemistry Chemical Physics, 12, 10896-10903, 10.1039/c0cp00402b, 2010.

---

## Author Comment (AC3)

**Response to Reviewer 3:**

This paper presents the hygroscopic behavior of BC particles generated from different types of fuel and aged with $SO_2$ for varying durations and explore the key factors that influence the hygroscopic of BC. They found that the presence of water-soluble substances in BC facilitates increase the number of water adsorption layers at high relative humidity. And the hygroscopicity of BC can be enhanced by the formation of sulfate ions due to heterogeneous oxidation of $SO_2$. I believe that the topic is interesting and it could be useful to the scientific community. However, some modifications are needed before they can be accepted.

**Reply:** We would like to thank you for the time and effort spent in reviewing the manuscript, which greatly improved the quality of our paper. The following is a point-by-point response to all the comments, and the manuscript has been revised substantially according to your comments.

**Major Comments**

**Q1.** The hygroscopicity of five different types of BC and the influencing factor was explored. However, only the aging of UBC was measured. The finding that low relative humidity has a limited effect on BC hygroscopicity is suitable for other black carbon (such as n-hexane flame BC, decane flame BC)? For example, will OC present in toluene flame BC lead to a decrease in the humidity turning point?

**A1.** We used UBC as a black carbon model for aging experiments because of sufficient sample amount for repeated experiments. Considering that the main product of $SO_2$ photooxidation on black carbon is sulfuric acid, we think that the weak effect of $SO_2$ ageing on the hygroscopicity of soot under low RH should be applicable to other black carbon. However, this need more experiments to verify, which is indeed our future plan to investigate how aging of OC affects the hygroscopicity of black carbon.

**Q2.** The key factor influencing the BC hygroscopicity was analyzed, including water-soluble ions, organic carbon content, and microstructure. However, BC size is very important for explore its hygroscopicity, but author seems to have neglected this. Please

clarify.

**A2.** We agree that BC size is very important for explore its hygroscopicity. We added the results of TEM experiments on various soot samples. TEM images of soot samples are shown below. All soot samples consisted of typical spherical particles, which formed long chainlike agglomerates as reported in other studies (Han et al., 2012; Liu et al., 2010). The diameter distribution of soot particles based on TEM analysis are also shown below. Particles exhibit a relatively uniform particle size distribution.

[Figure]

**Figure 2.** TEM images of n-hexane flame soot (A), decane flame soot (B), toluene flame soot (C), diesel soot (D), U-soot aggregates (E) before and (F) after aged with 5 ppm of SO$_2$ for 10 h.

[Figure]

**Figure 3.** Diameter distribution of n-hexane flame soot, decane flame soot, toluene flame soot and diesel soot and U-soot particles before and after aged with 5 ppm of SO2 for 10 h.

Nevertheless, we used the vapor sorption analyzer (VSA) to study the hygroscopicity of BC, which can only show the hygroscopic behavior of the bulk black carbon powder. Thus, the effect of particle size on the hygroscopicity of soot are not analyzed.

These discussions have been added in the revised manuscript (Line 154-167).

**Q3.** Why the microstructure of DBC and UBC was not measured?

**A3.** In section 3.1, the inorganic ions present in five different black carbons were analyzed using IC. The findings revealed that DBC and UBC contained significant amounts of $SO_4^{2-}$ and $NO_3^-$ ions. On the other hand, the three prepared black carbons exhibited minimal levels of inorganic ions. Thus, the three prepared black carbons were chosen to investigate the impact of factors other than inorganic ions on the hygroscopic properties of black carbon. Because inorganic components affect the hygroscopicity more significantly than OC and microstructure, the information of OC and microstructure of DBC and UBC was not shown.

**Q4.** The author has analyzed the influence of different factors on the hygroscopicity of BC, but in the ambient atmosphere, the main driving factor was difficult to identify. Could the author further discuss this through the results of this study?

**A4.** We agree that it is difficult to measure the hygroscopicity of black carbon in the atmosphere due to the complexity of its composition and atmospheric process. BC particles are emitted into the atmosphere from various sources, and their different origins lead to variations in their composition and microstructure, resulting in diverse hygroscopic properties. Moreover, the ambient atmosphere is highly intricate, comprising pollutants like nitrogen oxides, sulfur dioxide, volatile organic compounds, and others, which can alter the hygroscopic behavior of BC through heterogeneous reactions. Therefore, we intend to study the key properties that affect the hygroscopicity

of black carbon through experimental simulation. In our previous investigation, we found that the reaction between $SO_2$ and BC leads to the formation of a sulfuric acid coating on BC surfaces (Zhang et al., 2022). In this study, we further revealed the promoting effect of inorganic ions (especially $SO_4^{2-}$) generated by aging on hygroscopicity of BC. On the other hand, in previous study, the presence of water was found to enhance the oxidation of $SO_2$ on BC (He et al., 2020). In this study, we studied the key physicochemical properties that affect the hygroscopicity of fresh black carbon, which could be helpful for understanding the aging process of black carbon from different sources. In summary, the interaction between water and $SO_2$ on BC surfaces reinforces each other, highlighting the importance of considering the hygroscopic properties of BC in the atmospheric processes.

**Q5.** For the prepared BC, the Raman spectra suggested that the ID/IG of n-hexane BC, toluene BC and decane BC increased in turn, and the hygroscopicity of BC showed the same tendency. But it is not very valid to conclude that the ID/IG was positively correlated with hygroscopicity. The discussion should be expanded to support your conclusion.

**A5.** We have added the relevant content after careful consideration of your suggestion. Previous study found that the ratio ID/IG is inversely proportional to the graphite crystallite size $L_a$ (Knauer et al., 2009): $44/L_a = (I_D/I_G)$, where $L_a$ is the graphite crystallite size as determined by X-ray. The intensities of D and G bands have been widely determined using the sum of D1 and D4 bands and the sum of D2 and G bands, respectively. The result of Raman spectra demonstrates that there is a positive correlation between the ID4/IG of the three prepared soot samples and their hygroscopicity, while the $L_a$ of the three prepared soot samples exhibits a negative correlation with their hygroscopicity. These results imply that disordered graphitic lattice, polyenes, or ionic impurities (D4) could potentially serve as adsorption sites for water molecules. Moreover, smaller graphite crystallite size could enhance the adsorption capacity of water on soot.

These discussions have been added in the revised manuscript (Line 281-290)

**Specific Comments:**

**Q1.** 第 17 行"是"应改为"是"。

**A1.** Grammatical error has been corrected in the revised manuscript: In this work, the hygroscopic behavior of soot particles generated from different types of fuel combustion and aged with $SO_2$ for varying durations  was measured by a vapor sorption analyzer. (Line 15-17)

**Q2.** 284 行：作者发现随着老化时间的延长，UBC 上的硫酸根离子增加，而随着 $SO_2$ 老化，UBC 的 MRH 保持相对不变，这是否意味着硫酸盐涂层对低湿度下黑碳的吸湿性没有影响？

**A2.** We think so. According to our previous study, the sulfate detected on UBC could also exist in the form of sulfuric acid (Zhang et al., 2022). Moreover, IC results demonstrated that the amount of newly generated sulfate on UBC after 10 hours of aging was only 0.706 % of its original mass. Such a minute quantity of sulfuric acid formation may not exert a significant influence on UBC's water absorption characteristics within the range of 10-30% RH.

**Q3.** 302 行 您如何定义这种"高"相对湿度水平？这里应该指定"显着增强"和"高 RH"。

**A3.** Thanks for your suggestion. We have revised this sentence " Our results also demonstrated that a noticeable augmentation in the amount of water adsorbed on $SO_2$ aged BC at high relative humidity levels." to "Our results also demonstrated that a noticeable augmentation in the amount of water adsorbed on $SO_2$ aged soot at 90% RH, which is positively correlated with the amount of sulfate generated. (Line 317-319)". At 90 % RH, the adsorbed water layers on UBC increases with increasing aging times. By analyzing the relationship between sulfate production during $SO_2$ aging of UBC and its water absorption mass at 90% RH, we found a linear relationship with a high

correlation coefficient ($R^2$ = 0.9997). Our findings reveal that for every 1μg of $SO_4^{2-}$ produced on the surface of each milligram of black carbon, there is a corresponding increase of 1.82μg in its water absorption mass at 90% RH. These discussions have been added in the revised manuscript. (Line 300-304)

**Q4.** 第 320 行："并为提高我们的理解提供了基础............."，意思是提供什么....?

**A4.** Thanks for your reminding. We revised this sentence as "This study analyzed the key factors determining the hygroscopic property of soot, which can improve our understanding of the hygroscopic behavior of fresh soot and help to evaluate changes in hygroscopicity during the heterogeneous reactions of soot particles with pollutant gases in future studies." in the revised manuscript. (Line 335-337)

**REFERENCE**

Han, C., Liu, Y., Liu, C., Ma, J., and He, H.: Influence of combustion conditions on hydrophilic properties and microstructure of flame soot, J Phys Chem A, 116, 4129-4136, 10.1021/jp301041w, 2012.

He, G. Z. and He, H.: Water Promotes the Oxidation of SO2 by O-2 over Carbonaceous Aerosols, Environmental Science & Technology, 54, 7070-7077, 10.1021/acs.est.0c00021, 2020.

Knauer, M., Schuster, M. E., Su, D. S., Schlögl, R., Niessner, R., and Ivleva, N. P.: Soot Structure and Reactivity Analysis by Raman Microspectroscopy, Temperature-Programmed Oxidation, and High-Resolution Transmission Electron Microscopy, Journal of Physical Chemistry A, 113, 13871-13880, 10.1021/jp905639d, 2009.

Liu, Y., Liu, C., Ma, J., Ma, Q., and He, H.: Structural and hygroscopic changes of soot during heterogeneous reaction with O-3, Physical Chemistry Chemical Physics, 12, 10896-10903, 10.1039/c0cp00402b, 2010.

Zhang, P., Chen, T., Ma, Q., Chu, B., Wang, Y., Mu, Y., Yu, Y., and He, H.: Diesel soot photooxidation enhances the heterogeneous formation of H(2)SO(4), Nat Commun, 13, 5364, 10.1038/s41467-022-33120-3, 2022.

---

## Referee Report (RR1)

The author has discussed my problem in detail, and I recommend the article for further publication in ACP.